# Elevated Plasma D-Dimer Concentrations in Adults after an Outpatient-Treated COVID-19 Infection

**DOI:** 10.3390/v14112441

**Published:** 2022-11-03

**Authors:** Christa Meisinger, Inge Kirchberger, Tobias D. Warm, Alexander Hyhlik-Dürr, Yvonne Goßlau, Jakob Linseisen

**Affiliations:** 1Epidemiology, Faculty of Medicine, University Hospital of Augsburg, University of Augsburg, 86156 Augsburg, Germany; 2Institute for Medical Information Processing, Biometry and Epidemiology-IBE, LMU Munich, 80539 Munich, Germany; 3Vascular Surgery, Faculty of Medicine, University of Augsburg, 86159 Augsburg, Germany

**Keywords:** COVID-19, D-dimer, hypercoagulability, inflammation, outpatients

## Abstract

Elevated D-dimer plasma concentrations are common in hospitalized COVID-19 patients and are often associated with a worse prognosis, but it is not yet clear whether this also applies to outpatient cases. The present cross-sectional study evaluated D-dimer levels and their association with clinical parameters and inflammation biomarkers after a COVID-19 disease in individuals treated as outpatients. The study included 411 individuals (43.3% men) with an average age of 46.8 years (SD 15.2). Study participants who had acute COVID-19 disease at a median of 235 days (120; 323) ago were examined at the University Hospital Augsburg, Southern Germany, between 11/2020 and 05/2021. Plasma D-dimers were measured by a particle-enhanced immunoturbidimetric assay. Sixty-one subjects (15%) showed increased D-dimer concentrations (≥500 µg/L). Study participants with elevated D-dimer levels in comparison to subjects with levels in the reference range were significantly older, and more frequently reported a history of cardiovascular disease, hypertension, venous thromboembolism, and chronic venous insufficiency. In multivariable logistic regression analysis, CRP levels (OR 5.58 per mg/dL, 95% CI 1.77–17.60) and white blood cell count (OR 1.48 per nL, 95% CI 1.19–1.83) were significantly related to elevated D-dimers even after adjustment for multiple testing. However, acute or persistent symptoms were not significantly associated with increased D-dimers. Elevated D-dimer levels months after an acute COVID-19 disease seems to be associated with markers of inflammation. Further studies are needed to investigate the underlying pathophysiological mechanisms and consequences of prolonged D-dimer elevation in these patients.

## 1. Introduction

A number of previous studies have shown that patients with acute COVID-19 disease are in a state of hypercoagulability and thus have an increased risk of adverse thromboembolic events [1,2,3]. Elevated plasma levels of fibrin degradation D-dimers are commonly found in hospitalized COVID-19 patients and are associated with worse clinical outcomes [4], such as deep vein thrombosis and pulmonary embolism [5]. In a study from China [6] including 1099 patients, 43% of the non-severe patients showed raised plasma D-dimer levels, while this was the case in approximately 60% of intensive care unit (ICU) patients. There are a number of further studies showing similar results, where non-survivors developed significantly higher D-dimer concentrations compared to survivors at hospital admission [7,8,9].

Overall, it seems that increased D-dimers play a significant role in the context of a poorer prognosis in COVID-19, but the exact underlying pathophysiological mechanisms are not yet clear [9,10,11]. In addition to classic risk factors such as immobilization, smoking, and taking contraceptives, other COVID-19-specific factors or mechanisms may play a role with regard to increased thromboembolic risk. These include, for example, increased inflammation, hypoxia, and endothelial inflammation [12,13]. The binding of SARS-CoV-2 to the target host cell results in the release of inflammatory cytokines, associated with the migration of immune cells to the site of tissue damage [14]. 

These activated immune cells exacerbate endothelial injury through increased vascular leakage and microthrombus formation [15]. 

So far, investigations on blood coagulation parameters, in particular D-dimer levels, in post-COVID-19 patients treated as outpatients are scarce. Therefore, in the present study, clinical parameters together with coagulation and inflammation biomarkers were evaluated in post-COVID-19 patients treated as outpatients. 

## 2. Materials and Methods

The present study was conducted between 11/2020 and 05/2021 at the University Hospital Augsburg, Bavaria, Germany. The invitation to participate in the study was issued by the local health departments of the city and district of Augsburg. All registered residents with a positive SARS-CoV-2 smear by November 2020 (n = 1600) received a written invitation to participate in the voluntary study.

A total of 525 individuals, of which 463 participants were originally treated as outpatients and 62 participants as inpatients for COVID-19 disease, were examined. At the time of examination, at least 14 days had to have passed since the positive smear, and quarantine had to have ended. 

The study was approved by the Ethics Committee of the Ludwig-Maximilians-Universität, Munich (No. 20-735). Written informed consent was obtained from each study participant in accordance with the Declaration of Helsinki. 

In the present study, 463 participants (209 men and 254 women) with a history of COVID-19 disease treated as outpatients were included. Individuals who had already received vaccination against SARSCoV-2 before the examination (n = 26) and subjects with missing data regarding D-dimers (n = 26) were excluded, leaving 411 participants for the analyses (Appendix A). Due to missing values in the biomarker measurements, the analyses were based, in part, on different numbers. Reference is made to this accordingly. 

### 2.1. Data Collection 

Study participants underwent a physical examination with a focus on vascular complications, including ultrasonography of the legs, and were interviewed by means of a standardized face-to-face interview and a self-administered questionnaire regarding demographic data, weight and height, pre-existing conditions, medication use at the time of examination, and risk factors. Furthermore, the study participants were asked about symptoms/discomfort during the acute infection phase and during the 14 days prior to the study examination. Altogether, 42 symptoms or complaints were assessed: Increased temperature (37.5 to 38.0 degrees), fever (38.1 degrees and above), chills, cold or runny nose, nasal congestion, sore throat or throat pain, pain when swallowing, cough, hemoptysis, dyspnea or shortness of breath at rest, dyspnea or shortness of breath on exertion, feeling of pressure or chest pain, palpitations, heartburn, nausea or vomiting, abdominal pain, diarrhea, flatulence, loss of appetite, muscle or joint pain, muscle weakness, muscle stiffness, problems coordinating movements, feeling of pinpricks in the arms and legs, visual impairment, tearing eyes, red eyes or conjunctivitis, cyanosis, disturbance of the sense of smell, disturbance of the sense of taste, headache, vertigo, fatigue or exhaustion, sleepiness, sleep disorder, difficulties concentrating, memory impairment, depressed mood, anxiety or panic, mood swings, rash, and hair loss. Two new variables were created for every participant: One sum-score from the symptoms during the acute infection and a sum-score from the persistent symptoms. Finally, blood was drawn from study participants in a non-fasting state and analyzed immediately.

School education years were categorized into low (≤10 years of schooling) and high (>10 years of schooling). Marital status was categorized into married yes/no. A history of cardiovascular disease was defined as a history of myocardial infarction or stroke. In the assessment of pre-existing conditions and risk factors, the statement “I don’t know” was added to the statement “no.” The variable physical activity during the acute COVID-19 infection was recorded as 5 levels (significantly restricted, somewhat restricted, unchanged, somewhat increased, and significantly increased). A variable with 2 values was formed, where significantly restricted subjects were categorized as “restricted” and subjects reporting other values were categorized as “not restricted”. The Fatigue Assessment Scale (FAS), a reliable and valid tool, was used to assess fatigue [16]. In the present analysis, persons with scores above 21 were classified as having fatigue. 

### 2.2. Assessment of Blood Parameters 

Levels of anti-SARS-CoV2-spike IgG antibodies were measured by an enzyme-linked immunosorbent assay (ELISA) using the Elecsys immunoassay (Roche Diagnostics, Mannheim, Germany) following the manufacturer’s specifications. A positive result was indicated by a value > 0.4 U/mL and the upper measurement limit was set to 2500 U/mL. C-reactive protein (CRP) was measured in serum by an article-enhanced immunological turbidity test, where the aggregates are determined turbidimetrically (Cobas instrument, Roche Diagnostics, Mannheim, Germany). Interleukin-6 (IL-6) was analyzed in plasma samples using the Immunological ECLIA test (ElektroChemiLuminescence ImmunoAssay) on the Cobas instrument (Roche Diagnostics, Mannheim, Germany). Activated Partial Thromboplastin Time (aPTT; reference value: 26–36 s) was measured photometrically (Pathromtin SL, Siemens Healthcare); D-dimers (reference value: <500 µg/L) were analyzed in citrate plasma by a particle-enhanced immunoturbidimetric assay (Innovance D-Dimer Kit, Siemens Healthcare). Anti-β2glycoprotein IgG antibodies and anticardiolipin antibodies were measured by a quantitative ELISA (Euroimmun, Lübeck, Germany). Glucose concentrations were determined in serum using the GLUC3 assay on a Cobas c702 instrument (Roche Diagnostics GmbH, Mannheim, Germany). 

### 2.3. Statistical Analysis 

Quantitative data are reported as the median and interquartile range (IQR), and groups were compared by the Wilcoxon Rank Sum Test. Categorical data are presented as numbers and percentages and compared using Fisher’s exact test. To examine the association between the inflammation biomarkers, white blood cell count, CRP, and IL-6 and high D-dimer levels, logistic regression analyses were conducted. For each biomarker, a separate model was run and adjusted for age, sex, history of hypertension, CVD, diabetes, BMI, smoking status, marital status, school education, and time between acute infection and examination. As a sensitivity analysis, the models were recalculated after excluding all individuals who had a history of venous thromboembolism (VTE) before the acute COVID-19 disease. Finally, we performed regression analyses to investigate the relationship between the symptom sum-scores (acute infection and persistent symptoms), as well as selected symptoms at the time of acute illness or within the last two weeks before study participation, and high D-dimer levels. Adjustments were also made for the above-mentioned confounders. A *p*-value < 0.05 was considered statistically significant. To control the effect of multiple testing, we FDR-adjusted the obtained *p*-values. Statistical analysis was conducted using SAS version 9.4. 

## 3. Results

A total of 411 participants (178 males, 43.3%) were included in the study with a mean age of 46.8 (SD 15.2) years. Sixty-one people (15%) showed increased plasma D-dimer concentrations (≥500 µg/L) after a median of 255 days (137; 335) after the acute infection; of those, 17 individuals had even higher D-dimer values ≥ 1000 µg/L. 

Table 1 shows the sociodemographic and clinical characteristics and laboratory parameters in participants with and without elevated D-dimer levels. Subjects with elevated D-dimer levels were significantly older and more frequently married than subjects with D-dimer levels in the reference range. Furthermore, individuals with elevated D-dimer levels in comparison to persons without increased values more frequently reported a history of CVD, hypertension, venous thromboembolism, and chronic venous insufficiency.

Participants with high D-dimer levels in comparison to subjects with levels in the reference range showed significantly higher median values aPTT, white blood cell count, CRP, IL-6, anti-β2glycoprotein IgG antibodies, and SARS-CoV-2 IgG antibodies (Table 1).

Participants with D-dimer levels ≥1000 µg/L in comparison to subjects with D-dimer levels between >500 and <1000 µg/L did not differ significantly regarding clinical characteristics and laboratory parameters (Appendix A).

There was no linear correlation between CRP levels or white blood cell count and D-dimers. CRP levels out of range were found in 8 of the 61 (13.1%) subjects with high D-dimers, increased white blood cell count in 5/61 (8.2%) participants, elevated aPTT values in 2/61 (3.3%) subjects, and 1/61 (1.6%) study participants had decreased platelets. One individual with elevated D-dimer levels showed an IL-6 value outside the normal range. In the group with elevated D-dimer levels, 12 (19.7%) were positive for at least one antiphospholipid antibody; 10 subjects showed positive anti-β2glycoprotein IgM antibodies, 3 positive anticardiolipin IgM antibodies, and 1 positive anti-cardiolipin IgG antibodies. 

In Table 2, selected symptoms and complaints during the acute infection and within two weeks before study participation in subjects with and without elevated D-dimers are given. Subjects with elevated D-dimer levels in comparison to persons without increased values did not differ regarding shortness of breath at rest and on exertion, cough, and chest pain neither during the acute infection nor shortly before study participation. 

In multivariable logistic regression analysis, CRP levels were significantly related to elevated D-dimers. Furthermore, there was a significantly positive association between white blood cell count and increased D-dimer values independent of a variety of confounders (Table 3). 

In the sensitivity analysis, after excluding persons with a history of VTE before the COVID-19 infection, CRP and white blood cell count were still significantly associated with high D-dimer values with an OR of 5.07 (95% CI 1.74–14.78) and 1.29 (95% CI 1.04–1.58), respectively. 

To examine the relationship between selected symptoms during the acute infection as well as within two weeks before study participation and elevated D-dimers after several months, we conducted further logistic regression analyses. For these analyses, persons with a history of VTE before the COVID-19 infection were excluded. Shortness of breath at rest during acute infection, which might indicate pulmonary embolism, was not significantly associated with increased D-dimer levels in multivariable analysis (OR 1.36; 95% CI 0.72–2.56). Furthermore, shortness of breath two weeks before examination was not independently related to higher D-dimer levels (OR 1.05; 95% CI 0.36–3.07). In addition, symptoms such as dyspnea on exertion, cough, and chest pain during acute illness and within two weeks before examination showed no significant association with elevated D-dimers in multivariable analyses (Table 4). Furthermore, both symptom sum-scores were not significantly associated with elevated D-dimers in multivariable logistic regression analyses (acute infection: OR 1.00; 95% CI 0.96–1.04, *p*-value adj. 0.9243, and persistent symptoms: OR 0.97; 95% CI 0.92–1.03, *p*-value adj. 0.6668).

## 4. Discussion

This study demonstrated that 15% of men and women who recovered from COVID-19 disease treated as outpatients still had elevated plasma D-dimer levels several months later. Serum CRP concentrations and white blood cell counts were found to be independently related to elevated D-dimer levels. In contrast, no significant association was found with symptoms that might indicate VTE related to infection as the underlying cause of the elevation. There was also no independent relationship between the number of acute or persistent symptoms and elevated D-dimer concentration.

In our study, BMI, diabetes mellitus, and current smoking, i.e., known risk factors for severe COVID-19 that are also associated with a hypercoagulable state [17], did not differ between the two groups. This may be because younger (mean age 46.8 years, SD ± 15.2 years), healthier individuals with a history of mild COVID-19 disease participated in the present study. Overall, few comorbidities were present in the study participants. The proportion of married study participants differed significantly between the two groups. The reason for this difference is not yet clear. Whether this is a random result needs to be investigated in further studies. 

Prior investigations reported elevated D-dimer levels mainly in hospitalized, critically ill patients [18,19,20,21,22]. So far, studies on elevated D-dimers in exclusively outpatient-treated subjects months after the acute COVID-19 infection are rather scarce. A recent study of 150 convalescent COVID-19 patients with a mean age of 47.3 (SD 15.4) years (n = 81 outpatients, n = 69 inpatients) found that 29% of the patients treated exclusively as outpatients had elevated D-dimers up to four months after acute infection [23]. In the current study, in 15% of patients, after a rather mild COVID-19 disease, elevated D-dimer levels were found approximately eight months after the acute event. We performed ultrasonography to investigate the incidence of deep vein thrombosis in study participants, but the incidence of deep vein thrombosis (DVT) with two ultrasound-verified thromboses (unfortunately, D-dimer measurements were not available for these two cases) was very low and, thus, DVTs do not seem to be the cause of D-dimer elevation in this collective of initially mild COVID-19 cases. In addition, our study showed no association between disease symptoms that could indicate a history of pulmonary embolism and D-dimer elevation. However, recent studies could show that patients with prolonged SARS-CoV-2 symptoms are likely to have microthrombi, which we could not have identified by the ultrasound examination of the leg veins [24,25]. In addition, no CT pulmonary angiography was performed in our study to diagnose pulmonary embolism as a cause of D-dimer elevation. Previous studies reported that antiphospholipid antibodies, which are directed against various phospholipid-binding plasma proteins such as β2 glycoprotein I and prothrombin and bind phospholipids such as cardiolipin [26], were also found in severely ill COVID-19 patients [27]. They are known to be associated with infections, especially viral infections [28]. While some viruses, for example, hepatitis C, produce antiphospholipid antibodies that are associated with thrombotic diseases such as stroke, myocardial infarction, pulmonary embolism (PE), and DVT, antiphospholipid antibodies are usually discovered incidentally in viral infections [28,29]. In our study, the elevation of at least one antiphospholipid antibody was detected in almost 20% of individuals with elevated D-dimer levels; only two subjects in this group had two antiphospholipid antibodies. Whether, and if so, what role these antibodies play in the context of hypercoagulability in COVID-19 or whether they are also responsible for clot formation in milder cases should be the subject of further studies. There is evidence that the cytokine IL-6 plays an important role in the pathologic inflammatory response that triggers severe COVID-19, as high IL-6 concentrations in hospitalized COVID-19 patients are associated with a prognosis for more severe courses [30]. However, a specific mechanistic link with the development of thrombosis has not been established [31]. In our study, the levels of IL-6 in most of the participants were within the normal range; higher IL-6 levels were found in the group with elevated D-dimer levels compared with the group with D-dimer levels in the reference range. However, there was no significant association between IL-6 and high D-dimer values in multivariable analyses. 

CRP levels are known to be elevated in both acute-phase inflammation and chronic inflammatory diseases; it is produced in the liver in response to inflammatory cytokines, particularly IL-6. CRP levels were elevated in patients with COVID-19 and correlated positively with disease severity and mortality [31]. A meta-analysis of 20 studies that included 4843 patients with COVID-19 found a fourfold increased risk of poor outcomes in patients who had elevated CRP levels (>10 mg/L) [32]. In another study of 2782 patients with COVID-19, it was shown that more than 97% of patients had elevated CRP levels on admission; high baseline CRP levels were associated with progression to critical illness, acute kidney injury, VTE, and all-cause mortality [33]. Although the CRP values were within the normal range for almost all participants, our study showed a positive association between CRP serum concentrations and elevated D-dimer levels, even after adjustment for a number of confounders confirming the known link between inflammation and thrombosis in acute critical illness, including COVID-19 [34]. 

WBC count, a routinely measured parameter in clinical practice, is also a biomarker of chronic low-grade inflammation. Prior studies including patients with COVID-19 diseases treated in the hospital reported a relationship between higher WBC count at admission and disease severity [35]. In the present study based on a sample of outpatients, a significant association between WBC (although in the normal range) and higher D-dimer levels were found even after adjustment for a number of confounders. Our results are contrary to the findings of Townsend et al. [23] who observed no significant associations between elevated D-dimers and markers of inflammation in convalescent COVID-19 patients. The reasons for the different results are not clear, which is why further research on this topic is needed. 

In COVID-19, the most common coagulation/fibrinolytic abnormality detected is the increase in D-dimers, which is related to prognosis [36]. D-dimer is generated in the blood by the degradation of stabilized fibrin polymer by plasmin and is degraded again by the activation of fibrinolysis. The underlying pathophysiology leading to a hypercoagulable state may be related to a cytokine storm causing endothelial damage, microvascular thrombosis, and/or the development of prothrombotic antiphospholipid antibodies [36]. Recently, it was found that a high D-dimer level in COVID-19 survivors three months after hospital discharge was associated with the development of long COVID as a risk factor for pulmonary dysfunction [37,38]. The pathophysiologic mechanism underlying long COVID is not clear, but studies suggest that activation of the coagulation pathway, fibrinolysis, and pulmonary microvascular immunothrombosis may play a role during the onset of COVID-19 infection [39]. 

Because of the serious consequences of hypercoagulability, it is controversial whether anticoagulation should be considered in SARS-CoV-2 patients after the acute infection. In non-hospitalized SARS-CoV-2 patients, anticoagulation therapy should be initiated only when indicated [40]. In high-risk patients, such as those with malignancies or a history of venous thromboembolism, recent guidelines recommend that prophylactic anticoagulation with standard dosing should be considered [40]. In addition, treatment in affected patients could consist of discontinuing, for example, thrombosis-promoting drugs or dietary supplements. However, there is no general recommendation for anticoagulant treatment of COVID-19 patients after a mild disease. 

The present study has several limitations. First, we included mainly middle-aged men and women of German nationality, thus, generalizability to other age groups and ethnicities is limited. Second, only persons with a mild COVID-19 infection were included and, thus, the results are not transferrable to severe cases of infection. Third, CRP and not highly sensitive CRP values were available in this study. Furthermore, in 26 subjects, no D-dimer measurements could be conducted because no citrate plasma was available from these participants. Furthermore, no lupus anticoagulant was measured. Fourth, no information on blood parameters determined during acute infection was available that could have been considered in the analyses. Fifth, in our study, no computed tomography pulmonary angiography (CTPA) scans were performed to investigate whether pulmonary embolism was present in the participants. Sixth, the analysis was based on self-reported symptoms. As the participants with elevated D-dimers were older and more likely to have a history of CVD (among other conditions), the threshold for that group’s reported symptoms may have been different than the healthier group with normal D-dimer levels. Thus, recall bias cannot be excluded, in particular in the group with elevated D-dimer levels. Seventh, we do not have any information on whether the study participants had other infections close to the sample collection. Finally, no information on outcome parameters was available. 

In conclusion, the present results showed that elevated D-dimer levels are a common finding in outpatient-treated subjects with COVID-19 diseases even several months after the acute illness. It seems that elevated D-dimer levels might be related to inflammatory markers. Further studies are needed to elucidate the mechanisms underlying elevated D-dimers without a thromboembolic correlate after acute infection. Whether COVID-19-specific mechanisms or disease severity play a role in this context is still unknown and remains an important question to be answered. 

## Figures and Tables

**Table 1 viruses-14-02441-t001:** Characteristics of post-COVID-19 patients treated as outpatients (given as frequencies and percentages, or median and interquartile range), by elevated or normal-range plasma D-dimer concentrations.

	D-Dimer within Normal Range	D-Dimer Elevated	*p*-Value *
**Clinical characteristics**	n = 350	n = 61	
Sex (males)	155 (44.3)	23 (37.7)	0.4013
Age (years)	46 (33;56)	61 (50; 70)	<0.0001
School education (high)	238 (68.0)	40 (65.6)	0.7671
Marital status (yes)	213 (60.9)	47 (77.1)	0.0149
Body mass index (kg/m^2^) ^a^	24.9 (22.1; 28.4)	25.7 (22.5; 28.0)	0.6988
Hypertension (yes)	57 (16.3)	19 (31.2)	0.0112
Diabetes mellitus (yes)	11 (3.1)	4 (6.6)	0.2548
Depression (yes)	31 (8.9)	5 (8.2)	1.0000
Cardiovascular disease (yes)	15 (4.3)	8 (13.1)	0.0118
Venous thromboembolism before infection (yes)	8 (2.3)	6 (9.8)	0.0096
Chronic venous insufficiency (yes)	37 (10.6)	14 (23.0)	0.0110
Current smoker (yes)	123 (35.1)	28 (45.9)	0.1151
Post COVID-19 fatigue (yes) ^b^	137 (39.7)	22 (36.1)	0.6701
Time since acute infection (days)	222 (119; 322)	255 (137;335)	0.4980
Anticoagulation therapy (yes)	18 (5.1)	7 (11.5)	0.0760
**Laboratory parameters**			
White blood cell count (/nL) ^c^	6.40 (5.50; 7.31)	7.17 (5.92; 7.83)	0.0045
Hemoglobin (g/L) ^c^	140 (131; 150)	140 (131; 145)	0.3828
Platelets (/nL) ^c^	236 (206; 267)	223 (205; 269)	0.7692
Glucose (mg/dL) ^a^	87 (80; 99)	89 (79; 110)	0.1725
aPTT (s)	29 (28; 31)	28 (26; 31)	0.0028
C-reactive protein (mg/dL) ^a^	0.08 (0.06; 0.17)	0.12 (0.07; 0.32)	0.0010
IL-6 (pg/mL) ^e^	3.50 (2.50; 3.50)	3.50 (2.50; 3.89)	0.0005
D-dimer (µg/L)	199.5 (190.0; 285.0)	726.0 (617.0; 1094.0)	<0.0001
Anti-β2-glycoprotein IgG antibodies (U/mL) ^d^	2.2 (2.0; 2.9)	2.4 (2.0; 3.7)	0.0683
Anti-β2-glycoprotein IgM antibodies (U/mL) ^d^	4.4 (2.0; 9.9)	5.4 (2.0; 11.7)	0.2751
Anticardiolipin IgG antibodies (U/mL) ^d^	2.0 (2.0; 2.4)	2.0 (2.0; 2.3)	0.3675
Anticardiolipin IgM antibodies (U/mL) ^d^	2.0 (2.0; 2.8)	2.0 (2.0; 3.0)	0.3232
SARS-CoV-2 IgG antibodies (U/mL) ^f^	89.8 (25.8; 246)	169 (51.2; 349)	0.0156

^a^ based on 348 participants with D-dimer levels within the normal range; ^b^ based on 345 persons, ^c^ based on 346 persons, ^d^ based on 347 subjects, ^e^ based on 301 subjects, and ^f^ based on 349 subjects with D-dimer levels within the normal range; * Fisher’s exact test or Wilcoxon Rank Sum Test.

**Table 2 viruses-14-02441-t002:** Selected symptoms and complaints during the acute COVID-19 infection and within two weeks before study participation in subjects with and without elevated plasma D-dimer concentrations.

	D-Dimer within Normal Range	D-Dimer Elevated	*p*-Value *
	n = 350	n = 61	
**During acute infection**			
Calf pain ^a^	56 (16.1)	12 (19.7)	0.4612
Leg swelling ^a^	15 (4.3)	5 (8.2)	0.1985
Venous thromboembolism	3 (0.9)	0	
Shortness of breath at rest ^b^	109 (31.4)	25 (41.0)	0.1828
Dyspnea on exertion ^b^	198 (57.1)	37 (60.7)	0.6741
Cough ^a^	201 (57.8)	37 (60.7)	0.7786
Chest pain ^a^	131 (37.6)	23 (37.7)	1.0000
Restricted activity during acute infection	182 (52.0)	28 (45.9)	0.4071
**Within two weeks before study examinations**			
Shortness of breath at rest ^b^	25 (7.2)	8 (13.1)	0.1276
Dyspnea on exertion ^b^	84 (24.2)	20 (32.8)	0.1557
Cough ^a^	38 (10.9)	8 (13.1)	0.6598
Chest pain ^b^	38 (10.9)	5 (8.2)	0.6538

^a^ based on 348 participants with D-dimer levels within normal range; ^b^ based on 347 participants with D-dimer levels within normal range; * Fisher’s exact test.

**Table 3 viruses-14-02441-t003:** Association between circulating inflammatory markers and elevated D-dimer concentrations.

	Main Analysis			Sensitivity Analyses		
	OR (95% CI)	*p*-Value	*p*-Value Adj.	OR (95% CI)	*p*-Value	*p*-Value Adj.
CRP	5.58 (1.77–17.60)	0.0034	0.017	5.07 (1.74–14.78)	0.0029	0.017
IL-6	1.13 (1.02–1.26)	0.0219	0.0657	1.07 (0.93–1.23)	0.3358	0.6668
White blood cell count	1.48 (1.19–1.83)	0.0003	0.0045	1.29 (1.04–1.58)	0.0191	0.0657

Adjusted for age, sex, history of hypertension, cardiovascular disease, smoking status, history of diabetes, BMI, marital status, school education, and time between acute infection and examination.

**Table 4 viruses-14-02441-t004:** Association between selected symptoms/complaints and elevated plasma D-dimer levels (subjects with a diagnosis of venous thromboembolism before the COVID-19 disease were excluded).

	During Acute Infection			Within Two Weeks before Study Examinations		
	OR (95% CI)	*p*-Value	*p*-Value Adj.	OR (95% CI)	*p*-Value	*p*-Value Adj.
Shortness of breath at rest	1.36 (0.72–2.56)	0.3463	0.6668	1.05 (0.36–3.07)	0.9243	0.9243
Dyspnea on exertion	0.97 (0.51–1.83)	0.9187	0.9243	1.07 (0.54–2.14)	0.8416	0.9243
Cough	1.31 (0.70–2.48)	0.4001	0.6668	1.14 (0.42–3.11)	0.7950	0.9243
Leg swelling	1.49 (0.39–5.72)	0.5623	0.8435			

Adjusted for age, sex, history of hypertension, cardiovascular disease, smoking status, history of diabetes, BMI, marital status, school education, and time between acute infection and examination.

## Data Availability

The datasets generated during and/or analyzed during the current study are not publicly available due to data protection aspects but are available in an anonymized form from the corresponding author on reasonable request.

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
