# Peer review of "Elevated Plasma D-Dimer Concentrations in Adults after an Outpatient-Treated COVID-19 Infection"

_viruses, 2022, doi:10.3390/v14112441_

Round 1
Reviewer 1 Report
The authors investigated a relation of D-Dimer levels after COVID-19 infection with biographic data and selected inflammatory and coagulation parameters in a medium sized cross-sectional study. Elevated DD was found in 16.5% of patients. This finding was associated with elevated CRP and IL-6 levels. Other interesting findings are elevated age, history of VTE, chronic venous insufficiency, and cardiovascular diseases. SARS-CoV-2 IgG antibodies were also elevated in these patients compared to those with normal DD.
Statistical evaluation of the study can be improved by determination of specificity and related parameters.
The predictive value of DD alone and in combination with defined biographic and laboratory parameters. Results may be compared to evaluable prediction scores for outcome of COVID-19 patients.
A limitation of the study is also the lack of information on outcome parameters.
The extensive discussion on a relation of elevated DD in immunologic diseases and elevated laboratory infection can be substantially shortened.
Author Response
The authors investigated a relation of D-Dimer levels after COVID-19 infection with biographic data and selected inflammatory and coagulation parameters in a medium sized cross-sectional study. Elevated DD was found in 16.5% of patients. This finding was associated with elevated CRP and IL-6 levels. Other interesting findings are elevated age, history of VTE, chronic venous insufficiency, and cardiovascular diseases. SARS-CoV-2 IgG antibodies were also elevated in these patients compared to those with normal DD.
We thank the reviewer for his/ constructive comments, which were implemented to improve the manuscript.
Statistical evaluation of the study can be improved by determination of specificity and related parameters.
Answer: We would be happy to address this comment, but we do not understand it in its entirety. We therefore kindly ask the reviewer to specify the comment.
The predictive value of DD alone and in combination with defined biographic and laboratory parameters. Results may be compared to evaluable prediction scores for outcome of COVID-19 patients.
Answer: Thank you for this suggestion. To estimate the predictive value of DD alone and in combination with other parameters on COVID-19 outcomes is a nice idea. Unfortunately, we could not perform such an analysis because no outcome parameters were available in our study (see lines 349/350).
A limitation of the study is also the lack of information on outcome parameters.
Answer: We have added this limitation to the limitations section (see lines 349/350)
The extensive discussion on a relation of elevated DD in immunologic diseases and elevated laboratory infection can be substantially shortened.
Answer: As suggested, we shortened this part of the discussion (see lines 240-250).
Reviewer 2 Report
The presented manuscript is an interesting and well written study of d-dimer level in adults after an outpatient treatment of COVID-19.
In the materials and method section, the authors describe the inclusion and exclusion of study participants. I suggest adding a figure describing the number of participants across the inclusion and exclusion process with the detailed exclusion reasons to better enable the reader to grasp this process.
The study participants had their blood drawn and anti phospholipid antibodies level were measured. Why wasn’t the level of lupus anticoagulant measured (only anti-β2glycoprotein and anti-cardiolipin were measured)?
17 Individuals had elevated D-dimer level≥ 1000 μg/L (line 149). What were these patients’ characteristics (demographics, clinical, laboratory) in comparison to individuals with elevated D-dimer level≥ 500 μg/L but lower than 1000 μg/L? Can the author present this data using a table?
Table 1 and the results section describe that among patients with elevated d-dimer level there were significantly more frequently married individuals (line 152)- can the author assess what is the explanation?
Furthermore, well-known risk factors for inflammation and vascular damage, which are elevated BMI, diabetes mellitus and current smoking, did not differ between the study groups. Moreover, these are also known risk factors for severe covid-19 illness. The authors should discuss these findings and their potential explanation.
The authors found no correlation between d-dimer level and selected symptoms during acute covid-19 and 14 days before study participation. This is even more interesting considering that other inflammatory biomarkers (as CRP, WBC) were correlated to the elevated d-dimer level. So what is the authors’ advice to the clinician who encounters a patient several months after being treated as an outpatient for covid-19 and now presents for checkup with elevated d-dimer level, elevated CRP, WBC and maybe even an anti phospholipid antibody, but without any of the selected symptoms described at table 2? Should the patient undergo any further workup? Commence any kind of therapy? Repeat the lab work? Or just wait and see if any clinical signs develop? Please add a discussion of the matter.
Minor Comments:
Line 65: Instead of one continuous line, the phrase is stopped and changed to a new paragraph (line 66)- please correct.
Line 183: Instead of one continuous line, the phrase is stopped and changed to a new paragraph (line 189)- please correct.
Lines 302-304: there is a different font size compared to the rest of the text. Please correct.
Line 304: Instead of one continuous line, the phrase is stopped and changed to a new paragraph (line 305)- please correct.
Author Response
The presented manuscript is an interesting and well written study of d-dimer level in adults after an outpatient treatment of COVID-19.
First of all, we thank the reviewer for his/her comments. We feel that the manuscript has improved after implementing the suggestions.
In the materials and method section, the authors describe the inclusion and exclusion of study participants. I suggest adding a figure describing the number of participants across the inclusion and exclusion process with the detailed exclusion reasons to better enable the reader to grasp this process.
Answer: As suggested by the reviewer, we now provide a Flow-chart with the inclusions and exclusions of study participants (see online figure 1).
The study participants had their blood drawn and anti phospholipid antibodies level were measured. Why wasn’t the level of lupus anticoagulant measured (only anti-β2glycoprotein and anti-cardiolipin were measured)?
Answer: At the time of the study there were fresh indications of elevated anti-cardiolipin and anti-β2glycoprotein antibodies in patients with COVID-19, which is why we added them, but the significance was still completely unclear. Thus, the focus was not on autoimmune markers but on functional coagulation diagnostics. Therefore, no lupus anticoagulant was measured. We added this shortcoming in the limitations section (see line 337-339).
17 Individuals had elevated D-dimer level≥ 1000 μg/L (line 149). What were these patients’ characteristics (demographics, clinical, laboratory) in comparison to individuals with elevated D-dimer level≥ 500 μg/L but lower than 1000 μg/L? Can the author present this data using a table?
Answer: As suggested by the reviewer, we present the characteristics of the patients with D-dimer levels >1000 μg/L in comparison to patients with D-dimers >500 and <1000 μg/L in online table 1 in the supplements.
Table 1 and the results section describe that among patients with elevated d-dimer level there were significantly more frequently married individuals (line 152)- can the author assess what is the explanation?
Furthermore, well-known risk factors for inflammation and vascular damage, which are elevated BMI, diabetes mellitus and current smoking, did not differ between the study groups. Moreover, these are also known risk factors for severe covid-19 illness. The authors should discuss these findings and their potential explanation.
Answer: Thank you. As suggesed, we added some discussion on these points (see lines 232-239).
The authors found no correlation between d-dimer level and selected symptoms during acute covid-19 and 14 days before study participation. This is even more interesting considering that other inflammatory biomarkers (as CRP, WBC) were correlated to the elevated d-dimer level. So what is the authors’ advice to the clinician who encounters a patient several months after being treated as an outpatient for covid-19 and now presents for checkup with elevated d-dimer level, elevated CRP, WBC and maybe even an anti phospholipid antibody, but without any of the selected symptoms described at table 2? Should the patient undergo any further workup? Commence any kind of therapy? Repeat the lab work? Or just wait and see if any clinical signs develop? Please add a discussion of the matter.
Answer: Thank you for this comment. We now added a discussion regarding the treatment of patients in a hypercoaguable state after a mild disease (see lines 324-332).
Minor Comments:
Line 65: Instead of one continuous line, the phrase is stopped and changed to a new paragraph (line 66)- please correct.
Answer: we corrected it.
Line 183: Instead of one continuous line, the phrase is stopped and changed to a new paragraph (line 189)- please correct.
Answer: we corrected it.
Lines 302-304: there is a different font size compared to the rest of the text. Please correct.
Answer: we corrected it.
Line 304: Instead of one continuous line, the phrase is stopped and changed to a new paragraph (line 305)- please correct.
Answer: we corrected it.
Reviewer 3 Report
In this study, Meisinger and colleagues assessed D-dimer, blood cells, coagulation markers, and inflammatory markers in patients who previously had mild-to-moderate SARS-CoV-2 infection. They determined that ~15% of their population had elevated D-dimer, and that the patients with elevated D-dimer also had increased inflammatory markers. However, D-dimer did not correlate with self-reported respiratory and chest pain symptoms. The results are not surprising, as elevated D-dimer has been reported in other studies of SARS-CoV-2+ outpatients, and microthrombi have been reported in patients with Long COVID. There are several limitations with this study that the authors should address.
Major comments:
1. A major limitation of this study is that the authors have no data on the acute phase of infection, only analyzed a single timepoint post-infection, and that this timepoint varied considerably between study participants (ranging from two weeks to almost a year). I understand that the authors may not have access to plasma samples from the acute phase to perform measurements. However, they should at minimum analyze the effect of time on their results. For example, in the participants with elevated D-dimer, was D-dimer highest in samples collected closer to the initial infection? Or were the results independent of time? Either would be interesting. The timing also makes it very difficult to determine whether the elevated D-dimer and inflammatory markers are due to the initial covid infection. Do the authors have any information on whether study participants had other infections closer to the sample collection?
2. The authors are relying on self-reported symptoms for the analyses in tables 2 and 4. The paper would have been stronger if they had performed diagnostic assessments of respiratory and cardiac symptoms. As the participants with elevated D-dimer were older and more likely to have a history of cardiovascular disease (among other conditions), the threshold for that group reported symptoms may have been different than the healthier group with normal D-dimer. For example, a smoker who coughs regularly may be less likely to note coughing as a symptom, since it is part of their normal life experience.
3. The only exception to my comment above is that the authors used ultrasound to identify DVTs. They report that DVT incidence was not different between the high and low D-dimer groups, and use this to suggest that D-dimer is not related to thrombosis in this context. However, the work of the Pretorius group (for example, see Pretorius et al, Cardiovascular Diabetology 20(1): 172, 2021) indicates that patients with prolonged symptoms from SARS-CoV-2 infection are likely to have microthrombi, which would not be have been identified by ultrasound analysis of leg veins. Other studies have confirmed and extended these findings. The authors should discuss the possibility of microthrombi in their study subjects, instead of dismissing thrombosis. If they have sufficient sample available, they could also do their own microthrombi measurements. In addition, the authors did not look for thrombi within the lungs, the most common location that has been reported in acute patients.
4. The authors should assess whether CRP and white blood cell count correlate with D-dimer, or are simply elevated in the high D-dimer group (i.e. is there a linear correlation between D-dimer and CRP or WBC count in their population).
5. In the discussion (on page 7), the authors state “higher IL-6 levels were found in the group with elevated D-dimer levels…” They need to clarify that there was no significant difference in IL-6 once the data were adjusted for comorbidities.
Minor comments:
1. At the top of page 4, the text says that samples in the elevated D-dimer group were collected “after a median of 235 days (120; 323)". These numbers do not match the numbers in Table 1.
2. In the methods section, the authors state that 26 subjects were excluded for missing D-dimer measurements. I don’t understand this, as the authors did their own D-dimer measurements. Please explain.
3. There are several formatting errors, which made this manuscript more difficult to review. For example, on page 2, there is a line break after “positive SARS-CoV-2 smear by”, which makes it seem like the end of this sentence is missing. The next word (November) was moved to the start of a new paragraph. Similar issues appear throughout the text.
Author Response
In this study, Meisinger and colleagues assessed D-dimer, blood cells, coagulation markers, and inflammatory markers in patients who previously had mild-to-moderate SARS-CoV-2 infection. They determined that ~15% of their population had elevated D-dimer, and that the patients with elevated D-dimer also had increased inflammatory markers. However, D-dimer did not correlate with self-reported respiratory and chest pain symptoms. The results are not surprising, as elevated D-dimer has been reported in other studies of SARS-CoV-2+ outpatients, and microthrombi have been reported in patients with Long COVID. There are several limitations with this study that the authors should address.
We thank the reviewer for his/her good suggestions and revised the manuscript accordingly. We feel that the article has improved.
Major comments:
- A major limitation of this study is that the authors have no data on the acute phase of infection, only analyzed a single timepoint post-infection, and that this timepoint varied considerably between study participants (ranging from two weeks to almost a year). I understand that the authors may not have access to plasma samples from the acute phase to perform measurements. However, they should at minimum analyze the effect of time on their results. For example, in the participants with elevated D-dimer, was D-dimer highest in samples collected closer to the initial infection? Or were the results independent of time? Either would be interesting. The timing also makes it very difficult to determine whether the elevated D-dimer and inflammatory markers are due to the initial covid infection. Do the authors have any information on whether study participants had other infections closer to the sample collection?
Answer: As suggested by the reviewer, we now in addition adjusted in the analysis for the time between the acute infection and the examination. The results changed only marginally (see Tables 3 and 4).
We do not have any information on whether the study participants had other infections close to the sample collection. We added this limitation to the limitations section (see lines 348-349).
- The authors are relying on self-reported symptoms for the analyses in tables 2 and 4. The paper would have been stronger if they had performed diagnostic assessments of respiratory and cardiac symptoms. As the participants with elevated D-dimer were older and more likely to have a history of cardiovascular disease (among other conditions), the threshold for that group reported symptoms may have been different than the healthier group with normal D-dimer. For example, a smoker who coughs regularly may be less likely to note coughing as a symptom, since it is part of their normal life experience.
Answer: We totally agree with the reviewer. Self-reported data are indeed more unreliable than objectively measured data. Unfortunately, due to the limited COVID-19 research budget from the Bavarian government and the very limited time frame in which the study had to be conducted, it was not possible for us to perform extensive (instrumental) examinations of the study participants. Thus, recall bias cannot be excluded, in particular in the group with elevated D-dimer levels. We have added this important shortcoming to the limitations section (see lines 343-348).
- The only exception to my comment above is that the authors used ultrasound to identify DVTs. They report that DVT incidence was not different between the high and low D-dimer groups, and use this to suggest that D-dimer is not related to thrombosis in this context. However, the work of the Pretorius group (for example, see Pretorius et al, Cardiovascular Diabetology 20(1): 172, 2021) indicates that patients with prolonged symptoms from SARS-CoV-2 infection are likely to have microthrombi, which would not be have been identified by ultrasound analysis of leg veins. Other studies have confirmed and extended these findings. The authors should discuss the possibility of microthrombi in their study subjects, instead of dismissing thrombosis. If they have sufficient sample available, they could also do their own microthrombi measurements. In addition, the authors did not look for thrombi within the lungs, the most common location that has been reported in acute patients.
Answer: Thank you! We have now added some discussion on these points (see lines 263-266 and Ref. 24 and 25).
- The authors should assess whether CRP and white blood cell count correlate with D-dimer, or are simply elevated in the high D-dimer group (i.e. is there a linear correlation between D-dimer and CRP or WBC count in their population).
Answer: As suggested by the reviewer, we assessed whether there is a linear correlation between CRP and white blood cell count. This was not the case. We mentioned this in the results section (see line 171).
- In the discussion (on page 7), the authors state “higher IL-6 levels were found in the group with elevated D-dimer levels…” They need to clarify that there was no significant difference in IL-6 once the data were adjusted for comorbidities.
Answer: Thank you for this suggestion. We clarified that there was no significant association between IL-6 and high D-dimer values in multivariable analyses after excluding persons with a history of VTE before the COVID-19 infection (see lines 285-287).
Minor comments:
- At the top of page 4, the text says that samples in the elevated D-dimer group were collected “after a median of 235 days (120; 323)". These numbers do not match the numbers in Table 1.
Answer: Thank you for this hint. At the top of page 4 we mentioned the median follow-up time for the whole sample and not for the elevated D-dimer group. We now have corrected it so that the numbers match with the numbers in Table 1 (see line 150).
- In the methods section, the authors state that 26 subjects were excluded for missing D-dimer measurements. I don’t understand this, as the authors did their own D-dimer measurements. Please explain.
Answer: In 26 subjects no D-dimer measurements could be conducted because from these participants no citrate plasma was available. We added this shortcoming to the limitations section (see lines337-339).
- There are several formatting errors, which made this manuscript more difficult to review. For example, on page 2, there is a line break after “positive SARS-CoV-2 smear by”, which makes it seem like the end of this sentence is missing. The next word (November) was moved to the start of a new paragraph. Similar issues appear throughout the text.
Answer: Thank you for this hint. We now have corrected all this formatting errors.
Round 2
Reviewer 2 Report
The authors have addressed my previous comments in a satisfying manner.
However, there are few minor changes that ought to be made in the new version of the manuscript:
Line 324: The phrase begins with "eral limitations" instead of "several limitations". Please correct
Limitations section is repeated twice at the end of the text lines 324-340 and again lines 341-358. Please correct.
I strongly recommend that after correcting the manuscript text, the authors will read the text thoroughly prior to resubmitting it, as comments regarding the text editing after two previous submissions of the manuscript are inappropriate.
